Oceanic adults, coastal juveniles: tracking the habitat use of whale sharks off the Pacific coast of Mexico

Ramírez-Macías Dení tiburonballena@gmail.com 1
Queiroz Nuno 2 3
Pierce Simon J. 4
Humphries Nicolas E. 3
Sims David W. 3 5 6
Brunnschweiler Juerg M. 7
1 Tiburón Ballena México, Conciencia México , La Paz , Mexico
2 Centro de Investigação em Biodiversidade e Recursos Genéticos, Campus Agrário de Vairão, CIBIO/InBIO—Universidade do Porto , Porto , Portugal
3 The Laboratory, Marine Biological Association of the United Kingdom , Plymouth , United Kingdom
4 Marine Megafauna Foundation , Truckee , United States
5 Ocean and Earth Science, National Oceanography Centre Southampton, Waterfront Campus, University of Southampton , Southampton , United Kingdom
6 Center for Biological Sciences, Highfield Campus, University of Southampton , Southampton , United Kingdom
7 Independent Researcher , Zurich , Switzerland
Johnston David
Electronic publication date: 2017 May 4
Publication date: 2017
Volume: 5
Electronic Location ID: e3271
Received 2015 Jul 24; Accepted 2017 Apr 4
Copyright: ©2017 Ramírez-Macías et al.
Copyright year: 2017
Copyright holder: Ramírez-Macías et al.
License: This is an open access article distributed under the terms of the Creative Commons Attribution License, which permits unrestricted use, distribution, reproduction and adaptation in any medium and for any purpose provided that it is properly attributed. For attribution, the original author(s), title, publication source (PeerJ) and either DOI or URL of the article must be cited.
License URL: https://creativecommons.org/licenses/by/4.0/

Keywords: Movements, Habitat use, Diving behavior, Size segregation, Whale sharks, Pacific coast of Mexico, Habitat conservation

Funding: Save our Seas Foundation Project Aware Royal Geographical Society through (EXERCISE JURASSIC SHARK 2) A1 scuba, downtown aquarium Azul Marino Restaurant Palapas Ventana WWF-telcel Cabo Expeditions Shark Foundation and a private trust Fundação para a Ciência e Tecnologia (FCT) Investigator Fellowship IF/01611/2013 UK Natural Environment Research Council’s Oceans 2025 Strategic Research Programme (Theme 6 Science for Sustainable Marine Resources) Marine Biological Association (MBA) Senior Research Fellowship This project was supported by Save our Seas Foundation, Project Aware, Royal Geographical Society through (EXERCISE JURASSIC SHARK 2), A1 scuba, downtown aquarium, Azul Marino Restaurant, Palapas Ventana, WWF-telcel and Cabo Expeditions. SJP’s involvement in this study was supported by the GLC Charitable Trust, the Shark Foundation and a private trust. NQ was funded through a Fundação para a Ciência e Tecnologia (FCT) Investigator Fellowship (ref.: IF/01611/2013). Funding for NH was provided by the UK Natural Environment Research Council’s Oceans 2025 Strategic Research Programme (Theme 6 Science for Sustainable Marine Resources) in which DWS was a principal investigator. DWS was also supported by a Marine Biological Association (MBA) Senior Research Fellowship. The funders had no role in study design, data collection and analysis, decision to publish, or preparation of the manuscript.

==============================
Eight whale sharks tagged with pop-up satellite archival tags off the Gulf of California, Mexico, were tracked for periods of 14–134 days. Five of these sharks were adults, with four females visually assessed to be pregnant. At least for the periods they were tracked, juveniles remained in the Gulf of California while adults moved offshore into the eastern Pacific Ocean. We propose that parturition occurs in these offshore waters. Excluding two juveniles that remained in the shallow tagging area for the duration of tracking, all sharks spent 65 ± 20.7% (SD) of their time near the surface, even over deep water, often in association with frontal zones characterized by cool-water upwelling. While these six sharks all made dives into the meso- or bathypelagic zones, with two sharks reaching the maximum depth recordable by the tags (1285.8 m), time spent at these depths represented a small proportion of the overall tracks. Most deep dives (72.7%) took place during the day, particularly during the early morning and late afternoon. Pronounced habitat differences by ontogenetic stage suggest that adult whale sharks are less likely to frequent coastal waters after the onset of maturity.

Introduction

Electronic tagging studies have revolutionised our understanding of the behaviour and spatial ecology of marine animals. Linking these movement data to key environmental processes has led to critical insights into the movements of highly migratory marine vertebrates (Pade et al., 2009; Robinson et al., 2009; Block et al., 2011). This has supported the conservation of such species by provision of new information on where they range, which habitats are critical, what behaviours they perform, and why they may do so (Sims, 2010).

Whale sharks are highly mobile, with their movements driven by environmental conditions and associated biological productivity (Sequeira et al., 2013). Satellite tagging, in conjunction with remote sensing data, can be used to infer the influence of highly productive pelagic areas on the movement of these animals. Use of surface frontal zones in the open ocean represents a successful strategy to maximise prey encounter rates in patchy offshore seascapes, and has previously been documented in planktivorous sharks (Miller et al., 2015) and other pelagic shark species (Queiroz et al., 2016).

Most whale shark studies have taken place at their coastal feeding aggregations, where they typically exploit an ephemeral abundance of plankton or nekton in surface waters (Clark & Nelson, 1997; Ketchum, Galván-Magaña & Klimley, 2013; Robinson et al., 2013; Rohner et al., 2015a). Whale sharks also make deep dives, to a maximum-recorded depth of 1,888 m (Hueter, Tyminski & De la Parra, 2013), and appear to use meso- and bathypelagic depths regularly (Brunnschweiler et al., 2009). However, in almost all of the coastal locations where they are routinely sighted, the majority of sharks present are juvenile males (Rohner et al., 2015b). Few data are available for other ontogenetic stages (Rowat & Brooks, 2012), with the exception of large females tagged at Darwin Island in the Galapagos Islands, Ecuador (Hearn et al., 2016), and previously off the Baja Californian peninsula (Eckert & Stewart, 2001).

Whale sharks have been globally impacted by human activities, leading to an Endangered listing on the IUCN Red List (Pierce & Norman, 2016) and inclusion on Appendices II of both the Convention on International Trade in Endangered Species (CITES) and the Bonn Convention for the Conservation of Migratory Species of Wild Animals (CMS). The lack of information on whale shark habitat use is a significant knowledge gap, particularly with respect to the rare occurrence of the different ontogenetic stages and sexes within the same habitat or area. Frontal zones are effectively targeted by fisheries, which makes the study of pelagic movements in threatened species, like the whale shark, particularly relevant (Queiroz et al., 2016).

The Gulf of California (GoC) is a semi-enclosed sea known for its high biodiversity and primary production (Wilkinson et al., 2009). Globally, it is one of the only areas from which immature whale sharks, of both sexes, and large mature females have been routinely recorded (Clark & Nelson, 1997; Eckert & Stewart, 2001; Ramírez-Macías et al., 2007; Ramírez-Macías, Vázquez-Haikin & Vázquez-Juárez, 2012; Ketchum, Galván-Magaña & Klimley, 2013; Cochran et al., 2016). Some of the few reported neonatal whale sharks have also been found in this region (Wolfson, 1983). Together this identifies the region as one of only a few locations globally where the movements and habitat use of adult and juvenile whale sharks can be readily investigated in response to similar environmental factors.

Here, we used pop-off satellite-linked archival transmitter (PSAT) tags to examine the habitat use of whale sharks from the GoC, with a particular focus on adult sharks. In the only previous satellite tagging study on whale sharks from the GoC, Eckert & Stewart (2001) tracked adult females moving into the deep waters of the eastern Pacific Ocean. Our specific aims in the present study were to (i) determine if adult whale sharks are transient in the GoC, (ii) identify if juvenile whale sharks use different habitats compared to adult whale sharks, and (iii) investigate the influence of open-ocean frontal systems on their space use. Our results provide some of the first insights into the detailed horizontal and vertical movements of adult whale sharks, and provide information on the spatial ecology and behaviour of this threatened species in the eastern Pacific Ocean.

Figure 1 Study area and geolocated tracks of whale sharks.

(A) Map of the study area in the north-east Pacific showing main locations mentioned in the manuscript, BG, Banco Gordo; (B) Geolocated tracks of whale sharks in the north-east Pacific Ocean overlaid on bathymetry; stars denote pop-off locations and red square the area enlarged in (A).

Materials & Methods

This research was carried out under the general auspices of CONACYT (Consejo Nacional de Ciencia y Tecnología), DGVS (Dirección General de Vida Silvestre), SEMARNAT (Secretaría del Medio Ambiente y Recursos Naturales) and CONANP (Comisión Natural de Áreas Naturales Protegidas). These are the relevant Mexican authorities governing all research actions on wildlife and protected animals and areas in Mexico. CONACYT registration: RENIECYT No. 030 (currently 1602199) and 13920. DGVS authorization numbers are: SGPA/DGVS/02677/08, SGPA/DGVS/02888/09, SGPA/DGVS/03848/10, SGPA/DGVS/031 55/11, SGPA/DGVS/03362/12.

Table 1 Tagging details and performance of PSATs attached to whale sharks in the Gulf of California.

Set pop-off interval is the programmed time PSATs were schedule to report data after deployment. * denotes PSATs attached to whale sharks (WS) that did not uplink to the Argos satellite system. J, juvenile; A, adult; M, male; F, female; cp, constant pressure; td, too deep emergency release; ukn, unknown; DAL, days-at-liberty; nr, nonreporting.

Shark ID	Sex	Estimated length (m)	Maturity status	Tagging date	Set pop-off interval (days)	Pop-off date	Pop-off area	Pop-off reason	DAL	Pressure data availablea/usedb for analysis (%)	Temperature data availablea/usedb for analysis (%)	
				2008								
JM1	M	7	Juvenile	19 November	274	13 December	Bahía de La Paz	cp	25	100/100	100/100	
				2009								
*WS2	M	7	Juvenile	17 January	274				nr			
*WS3	M	7	Juvenile	17 January	274				nr			
*WS4	F	12	Pregnant	23 May	365				nr			
AF1	F	12	Pregnant	23 May	365	6 July	offshore	cp	44	100/99.7	98.9/95.9	
AF2	F	10.5	Pregnant	24 May	365	5c October	offshore	uknd	134	30/30	30.4/29.4	
*WS7	F	11.5	Pregnant	26 May	365				nr			
				2010								
JM2	M	7	Juvenile	10 March	274	23 March	Bahía de La Paz	cp	14	100/100	100/100	
*WS9	M	5	Juvenile	11 March	365				nr			
JF1	F	5	Juvenile	13 March	365	9 May	Bahía de Los Angeles	td	58	99.5/98.1	99.9/97.5	
*WS11	F	12	Pregnant	2 June	365				nr			
*WS12	F	11	Pregnant	23 June	365				nr			
AF3	F	11.5	Pregnant	29 June	365	24 September	offshore	ukn	88	98.4/98.2	96.5/92.5	
AM1	M	9	adult	13 July	365	21 August	offshore	tde	40	75.9/74.3	76.9/59.1	
AF4	F	11	Pregnant	13 July	365	13 September	offshore	ukn	63	97.2/96.9	95.9/92.9	
				2011								
*WS16	M	6	Juvenile	16 February	274				nr			
				2012								
*WS17	F	12	Pregnant	8 May	365				nr			
Notes.

a Including delta limited values.

b Excluding delta limited values.

c The exact pop-off date is unknown. Constant 0 m depth readings were archived as of 11 October. No archived depth data were transmitted for after 5 October, so this date was considered pop-off date although it is possible that the tag stayed on for more days.

d The constant pressure release mechanism was activated on 21 September, but the PSAT stayed attached to the whale shark until at least 5 October. It is possible that the popoff wire did not fully corrode and it took some time before the PSAT was able to break the weakened wire.

e The too deep emergency release mechanism was activated on 11 August, but the PSAT stayed attached to the whale shark until 21 August when it detached and came to the surface. It is possible that the popoff wire did not fully corrode and it took some time before the PSAT was able to break the weakened wire.

Study site and electronic tagging

Seventeen whale sharks, ranging from 5 to 12 m total length, were tagged with PSATs (PTT-100 standard rate PSAT; Microwave Telemetry Inc.): 16 between 2008 and 2011, in inshore waters of Bahía de La Paz, the marine protected area (MPA) of Espíritu Santo Island (ESI), and a single individual at Banco Gordo in 2012 (Fig. 1A; Table 1). A spotter airplane was used to locate sharks off ESI. Flights followed pre-defined transects over a duration of three hours. The east and west coasts of ESI (including El Bajo), respectively, were covered over consecutive days. Surveys were conducted weekly in May 2009 and from May to July in 2010. Whale shark positions were communicated to the research vessel (7 m) via radio, allowing the vessel to conduct in-water studies. Shark lengths were measured by a swimmer using a metric band or estimated by repeated on-board observations of the whale shark swimming parallel to the research vessel (Ramírez-Macías, Vázquez-Haikin & Vázquez-Juárez, 2012; Fig. S1A). Sex of the animal was determined by the presence of claspers on males, which are visible from birth. Clasper morphology was used to distinguish juvenile from adult males: claspers are short, soft, and smooth in sexually immature males, but quickly grow and calcify during maturation (Norman & Stevens, 2007). Females were categorized as either juvenile or adult based on their estimated lengths and external indications of pregnancy (i.e., distended abdomen; Ramírez-Macías, Vázquez-Haikin & Vázquez-Juárez, 2012; Acuña Marrero et al., 2014; Robinson et al., 2016). While we acknowledge the difficulty of confirming pregnancy in adult female sharks, the visual assessments made for the present study were based on extensive field observations of adult sharks, both pregnant and non-pregnant, by the first author (Ramírez-Macías, Vázquez-Haikin & Vázquez-Juárez, 2012). Given the uniquely distended nature of the pelvic region in these sharks, pregnancy seems to be the most parsimonious explanation. However, given that pregnancy was not possible to verify by direct methods (e.g., ultrasonography) we refer to putative pregnancy in whale sharks in this study. We have proceeded on that assumption. The flank of each whale shark was also photographed for individual identification (Marshall & Pierce, 2012; Ramírez-Macías, Vázquez-Haikin & Vázquez-Juárez, 2012).

Whale sharks were tagged using a spear gun with a standard rubber band. The full tag setup consisted of the PSAT unit, a 136 kg test monofilament tether and a stainless steel tag anchor. All whale sharks were tagged on the left side at the base of the first dorsal fin. Tags were programmed to pop off after deployment intervals of either 9 (274 days) or 12 (365 days) months (Table 1). During deployment on the study animal, Microwave Telemetry’s standard rate PSATs record temperature, depth and light-level every 2 min (for a detailed description on how PTT-100 standard rate PSATs record, archive and transmit data see Brunnschweiler (2014) and http://www.microwavetelemetry.com/fish). The maximum depth the tag model used in this study can archive is 1,285.7 m. However, an emergency release feature detaches the tag automatically when the shark is deeper than ∼1,250 m for more than 15 min to prevent the tag from being crushed at depth. In all tags attached to adult whale sharks, and juveniles JM1 and JM2 (Table 1), a constant pressure release feature, set at ±10 m for 4 days, was enabled (Brunnschweiler et al., 2009; Brunnschweiler, Queiroz & Sims, 2010). In all other tags attached to juvenile whale sharks (Table 1), the constant pressure release feature was set at ±3 m for 4 days.

Data analysis

Eight of the 17 PSATs attached to whale sharks (47.1%) uplinked to the Argos satellite system and transmitted data (Table 1). Pop-off date and the reasons for detachment were determined by constant 0 m depth readings, start time of transmission to the Argos satellite system, and status information transmitted by the PSAT. After pop-off, PSATs AF2, AF3 and AF4 each floated on the surface for several days (6, 10, and 10 respectively) before uplink to the Argos satellites, and therefore true pop-off positions are unknown.

Whale shark locations were estimated using satellite-relayed data from each tag. Recovered light-level data were used by Microwave Telemetry (MT) to estimate local time of midnight or midday for longitude calculations and day length for latitude calculations using a proprietary algorithm derived from standard celestial algorithms. Light-derived position estimates were subsequently improved by combining: (i) a swim speed filter (to constrain distances), (ii) remotely sensed SST data and tag recorded SST values to constrain probable locations, and (iii) location bathymetry and tag recorded depths to filter positions that would be too shallow. To achieve this, the MT-estimated latitudinal and longitudinal positions and error fields were used to define the area that contained all the possible positions for a given location. A 1 m s−1 swim-speed filter (which is consistent with cruise speeds of pelagic sharks (Gunn et al., 1999; Sims, 2000)) was used for each estimated position to generate a circular area representing the possible positions that could be reached by the shark in the time between the current and the next location estimate. The intersection of the swim speed area and the error field produced a sub-set of possible points which were then filtered to ensure the maximum dive depth did not exceed that which was possible given the bathymetry at that location. All the available filtered points were then checked against the corresponding SST map (using OSTIA high-resolution, spatially complete, global coverage remote-sensing images) for that day and any points within a 2 °C threshold between the tag-recorded temperature and the SST were recorded as possible ‘waypoints’. After all locations were processed, the waypoints at each location were scored according to the difference in SST and distance from the original estimated location with high scores representing poorer SST matches and longer distances. The best (lowest) scoring waypoints were then connected to form a most probable track. Resighting data were also used when available by fitting them as known locations during the filtering process and these were incorporated into the final tracks.

Gaps between consecutive dates were linearly interpolated to one position per day to obtain unbiased estimates of shark space use. Furthermore, to account for the spatial error around individual geolocations, these were randomly resampled 100 times (point density data) along previously reported tag-specific longitudinal and latitudinal Gaussian error fields, 0.16°in longitude and 1.19°in latitude (Hueter, Tyminski & De la Parra, 2013; Fig. S2). Resampled geolocations were then combined with satellite-derived environmental data; the environmental data used were daily (i) sea surface temperature, SST and from NOAA Optimum Interpolation Quarter Degree Daily SST Analysis (OISST) data. Based on the OISST data we also calculated (ii) daily SST maximum gradient maps by calculating, for each pixel, a geodetic—distance-corrected maximum thermal gradient (°C/100 km), and (iii) monthly merged chlorophyll a levels (0.25°spatial resolution), acquired from GlobColour (European Space Agency—ESA). Before further analysis, point density and environmental data were averaged into 0.5°grid cells (Fig. S3).

Point density data were used to calculate shark space use by performing a kernel density interpolation with barriers in ESRI ArcGIS [v. 10.3]. To analyse the spatial relationship between environmental variables and shark space-use, a Generalized Linear Mixed Model (GLMM) analysis with penalized Quasi-Likelihood parameter estimation (PQL; to account for non-normal error distributions) was employed using R (Venables & Ripley, 2002; Austin et al., 2006). In the model, shark space-use was set as a random factor, while environmental parameters were set as fixed effect factors. An autocorrelation structure of order 1 (corAR1) was used to account for the temporal correlation in the dataset (Zuur et al., 2009). GLMM models were fitted with a normal (Gaussian family) distribution. Finally, to evaluate model performance, the concordance index (C-index; Harrell et al., 1984) was calculated using R (Hmisc package). The C-index estimates the probability of concordance between predicted and observed responses, varying between 0.5 and 1.0 with the following classification: excellent if above 0.9; good 0.9–0.8; reasonable 0.8–0.7; poor 0.7–0.6 and unsuccessful 0.6–0.5 (Swets, 1988). Model results are given in the following format: β ± SD, P, C-index, where β is a measure of the slope of the relationship. Presence of sharks in ‘coastal’ waters is defined here as continental shelf (<200 m), while ‘offshore’ refers to the open ocean (>200 m depth).

Vertical movements and diel patterns in behaviour were analyzed for individual whale sharks using archived time–depth and time–temperature data (depth and temperature resolution = ∼5.4 m and ∼0.17 °C, respectively). Due to the non-Gaussian distribution of the data, mean and median were used to summarize the results, and the non-parametric Mann–Whitney U test for comparisons (significance level = 0.05). Except for the analysis of deep dives (see below), delta limited depth and temperature values (Brunnschweiler, 2014) were removed from the raw datasets, with mean and medians used to summarize the results. Vertical and thermal niches were determined using daily minimum/maximum depth and temperature values recorded at 2 min intervals (Brunnschweiler, 2014).

Archived and daily minimum/maximum depth readings were assigned to one of three categories: epipelagic (0 m, <200 m), mesopelagic (>200 m, <1,000 m) and bathypelagic (>1,000 m). Pressure readings of 0 and 5.4 m were defined as the whale shark being at the water surface (Brunnschweiler & Sims, 2012) and deep diving was defined as diving to meso- or bathypelagic depths. To investigate at what time(s) of the day deep diving occurred, all archived depth readings including delta limited values in the meso- or bathypelagic zones were assigned to one of 24 hourly bins. An individual deep dive was defined as the time between the whale shark descending from <50 m to meso- or bathypelagic depths until reaching the surface again. The time-depth series (archived depth values at 15 min intervals) of individual deep dives with maximum non-delta limited archived depth recordings in the meso- and bathypelagic zones were plotted, visually classified based on their time-depth profiles, and their dive geometry characterized following definitions provided by Gleiss, Norman & Wilson (2011). To test the hypothesis that deeper dives (maximum daily depth) occurred in less productive waters, a Spearman Rank correlation was performed (data not normally distributed; Shapiro–Wilk; W = 0.588, p < 0.001) between maximum daily depth (using pooled data for all tracked whale sharks) and mean chlorophyll-a concentration.

Results

Resightings and tag performance

All eight PSATs prematurely detached from sharks after being attached for 14–134 days (Table 1). Of these, four (JM1, AF1, AF2, JM2) and two (JF1, AM1) PSATs released due to activation of the constant pressure or the crush depth emergency release mechanisms, respectively. For two PSATs (AF3, AF4), the reason for premature release was unknown (Table 1). Except for PSAT AF2, for which three days were missing, latitude and longitude positions were transmitted for all days of the respective tracks. PSATs transmitted between 30 and 100% of archived pressure and temperature data (Table 1). Data from six PSATs (AF1, AF2, JF1, AF3, AM1, AF4) contained between 0.1 (AF2) and 2% (AM1), and 2.5 (JF1) and 23.2% (AM1) delta limited depth and temperature values, respectively.

Eight whale sharks were resighted post-tagging using photo-identification (Table 2). Six of them were resighted in the coastal waters of Bahía de La Paz (Fig. 1A). Two whale sharks were resighted away from the tagging site. The juvenile male tagged in Bahía de La Paz on 16 February 2011 was resighted, without the PSAT, in the protected area of Bahía de Los Angeles on 13 August 2011. The pregnant whale shark tagged at Banco Gordo on 8 May 2012 was resighted, without the tag, in the protected area of Roca Partida, the smallest of the four Revillagigedo Islands (Fig. 1B), on 22 November 2012, and was reported to be pregnant.

Table 2 Resighting history of eight whale sharks.

Shark ID	Number of resightings	Location of resighting (s)	Min/max time (days) after tagging date	Remarks	
JM1	1	Bahía de La Paz	28/28	Monofilament tether still in place.	
JM2	3	Bahía de La Paz	1/16	PSAT attached at first two resightings; only monofilament tether at last resighting.	
JF1	4	Bahía de La Paz	9/272	Resighted with PSAT attached 9, 26 and 28 days after tagging; without PSAT at last resighting.	
WS2	4	Bahía de La Paz	12/53	PSAT attached at first three resightings; only monofilament tether at last resighting.	
WS3	15	Bahía de La Paz	3/292	Resighted 14 times with the PSAT attached until 65 days after tagging; only monofilament tether at last resighting.	
WS9	2	Bahía de La Paz	21/286	First resighting with, last resightin without PSAT attached.	
WS16	1	Bahía de Los Angeles	178/178	Without PSAT attached.	
WS17	1	Revillagigedo Islands	198/198	Without PSAT attached.	

Horizontal movements

There was a marked difference in the horizontal movements exhibited by juvenile and adult whale sharks. All juveniles remained in the GoC, while all adult animals moved out into the eastern Pacific Ocean. The two juvenile males JM1 and JM2, tagged for 25 and 14 days respectively, remained in the shallow waters of Bahía de La Paz where they were tagged. All six other whale sharks left the tagging area and moved large distances (Fig. 1B). Tagged animals mainly occupied two areas, one in the coastal waters around the southern part of the Baja California peninsula, and the other offshore in the eastern Pacific Ocean (Fig. 2).

Figure 2 Major areas of residency.

Kernel density plot (estimated using the coastline as a barrier) showing the two major areas of prolonged residency (in number of days), one inside the Gulf of California and another offshore; dashed line represents 95% isopleth.

Juvenile JF1 spent the first weeks after tagging in and north of Bahía de La Paz before moving north to 28.7°N–113.0°W where the tag popped-off south of Bahía de Los Angeles, on 9 May, after 58 days (Fig. 1B). Pregnant whale sharks AF1 and AF2 both left the tagging area, and after a week the GoC, moved to the same offshore area (Fig. 1B). It appears that both animals moved clockwise in a large circle in June/July 2009 before heading north. Whereas shark AF1 lost its tag after 44 days on 6 July, shark AF2 was tracked for another three months. In that time period, it continued to move north (Fig. 1B). All three females, adults AF1 and AF2 as well as juvenile JF1, remained in productive waters for longer time periods before moving north.

Putative pregnant whale sharks AF3 and AF4, tagged in 2010, left the GoC shortly after tagging, similar to sharks AF1 and AF2 from 2009. The 2010 sharks remained associated with productive coastal waters in July/August, before moving to an oligotrophic offshore area in September where tags popped off (Fig. 1B). Similar to the pregnant females, the only adult male (AM1) tagged in this study left the GoC within a week after tagging and quickly moved south where it stayed in a relatively confined area in August until the PSAT popped off (Fig. 1B).

GLMM analysis showed that shark space-use was significantly influenced by SST and SST gradients (fronts). However, temperature had a negative (albeit weak) effect on space-use, while stronger thermal gradients had a positive effect (Table 3) indicating whale sharks tended to spend more time in frontal regions linked with upwelling (which resulted in colder surface water). No significant relationship was found between shark space-use and chlorophyll a concentration and model performance was reasonable (C-index = 0.7).

Table 3 Results of the GLMM model.

Model results are given in the following format: β ± SD, P, C-index, where β is a measure of the slope of the relationship and C-index the concordance index between predicted and observed responses.

Variable	β± SD	P	
Chlorophyll a concentration	−0.28 ± 0.15	0.06	
SST	−0.27 ± 0.03	<0.001*	
SST gradients	25.57 ± 7.14	<0.001*	
Notes.

* denotes significant relationship.

Vertical movements

Mean depth and temperature experienced by individual whale sharks ranged between 1.1 (JM2) and 29.4 m (AM1), and 21.64 (AF1) and 25.48 °C (JM1), respectively (Table 4). Significant day and night differences in mean depth were detected in all but one individual (JM1). Mean depth was greater during the night for all sharks except for JF1, the one juvenile that moved significantly away from the tagging area, which stayed deeper during the day (Table 4).

Table 4 Summary statistics.

Depth (m) and temperature (°C) summary statistics for whale sharks tagged in the Gulf of California.

Shark ID	Depth (m)	Temperature (°C)	
	Mean ± SD	Median	Mean ± SD day	Median day	Mean ± SD night	Median night	Maximum	Mean ± SD	Median	
JM1	3.9 ± 4.8	5.4	3.7 ± 4.5	5.4	4.2 ± 5.1	5.4	32.3	25.48 ± 0.73	25.48	
AF1	9 ± 14.6	5.4	7.5 ± 16.4	5.4	10.6 ± 12.5	5.4	882.3	21.64 ± 1.52	21.70	
AF2	13.4 ± 12.6	10.8	12.5 ± 12.7	10.8	14.4 ± 12.4	10.8	1076	22.17 ± 1.74	22.20	
JM2	1.1 ± 2.7	0	0.5 ± 1.9	0	1.7 ± 3.2	0	21.5	22.29 ± 0.49	22.20	
JF1	6 ± 27.4	0	8.1 ± 34.7	0	3.9 ± 17.3	0	1285.8	22.44 ± 1.77	22.71	
AF3	6.3 ± 16	0	4.9 ± 16.1	0	7.8 ± 15.8	0	607.9	22.62 ± 1.67	22.71	
AM1	29.4 ± 57.2	21.5	25.1 ± 41.9	5.4	33.1 ± 68.4	26.9	1285.8	24.53 ± 3.86	25.66	
AF4	7.7 ± 16.4	0	6.3 ± 15.3	0	9.1 ± 17.2	5.4	919	23.58 ± 1.99	23.73	

Excluding juvenile whale sharks JM1 and JM2, which did not leave the tagging area over the duration of PSAT attachment and dived to relatively shallow maximum depths of 32.3 and 21.5 m, respectively, time spent at the surface (<6 m) constituted 65 ± 20.7% of total time for the other whale sharks. These six whale sharks, which all moved long distances from the area where they were tagged (Fig. 1B), spent most of their time in 20–26 °C water. All exhibited regular diving throughout their tracks and had broad vertical and thermal niches. Three individuals were recorded in each of the mesopelagic (AF1, AF3, AF4) and bathypelagic (AF2, JF1, AM1) zones several times during PSAT attachment (Fig. 3). Two sharks, JF1 and AM1, recorded dives to or exceeding the maximum depth that the tags could record (1,285.8 m; Table 4).

Figure 3 Diving activity.

Time-depth series recorded by tags attached to whale sharks leaving Bahía de La Paz (tagging area) at large scale. Dots denote daily maximum depths (epipelagic: 0 m, <200 m (yellow); mesopelagic: >200 m, <1000 m (green); bathypelagic: >1000 m (red) zones).

Individual differences with regards to overall diving behaviour were evident. The four putative pregnant whale sharks (AF1, AF2, AF3, AF4) that left the GoC showed similar diving patterns, with maximum daily depths in the epipelagic zones on most days (Figs. 3A, 3B, 3D and 3F). The juvenile whale shark (JF1) that stayed within the GoC stayed shallow with a maximum diving depth of 32.3 m for the first 49 days of its track, excluding 3 April 2010 (Fig. 3C). Then, for the remaining nine days before the PSAT popped up due to activation of the deep-depth release mechanism, this whale shark showed increased deep diving activity into the meso- and bathypelagic zones during its northwards movement to the upper GoC (Figs. 1B and 3C). This shark (JF1), with a mean depth of 6 m over the entire duration of the track, marked the shallowest end of the spectrum. The adult male whale shark AM1 was the individual with the deepest mean depth (Table 4). This whale shark quickly left the GoC and moved south (Fig. 1B), with daily maximum depths in the meso- and bathypelagic zones on 52.5% of days tracked (Fig. 3E).

Meso- and bathypelagic diving and dive geometry

Overall, sharks spent a relatively small amount of time in the meso- and bathypelagic zones (Fig. 3). Dives to maximum depths in these zones occurred largely during the day, with 72.7% of all >200 m recordings archived between 06:00 and 18:00 (Fig. 4). Most deep diving activity was performed in the morning between 07:00 and 09:00 (28.6%) and in the late afternoon between 15:00 and 18:00 (20.8%) (Fig. 4).

Figure 4 Meso- and bathypelagic diving.

Percent archived depth recordings in the meso- and bathypelagic zones at hourly intervals from all whale sharks leaving Bahía de La Paz (tagging area) at large scale. Note that delta-limited dives were also included. The total number of depth recordings was 154: mesopelagic (M; green = 43), bathypelagic (B; red = 4) and delta limited (DL; dark grey = 107). Grey shaded areas denote night.

Fifteen dive profiles with maximum depth readings in the mesopelagic zone were available for visual classification. Nine contained delta limited depth values (Brunnschweiler, 2014). Three were characterized as isolated V-dives (Fig. 5A), two as V-dives in series (Fig. 5B), four as U-dives (Fig. 5C), and six dive profiles could not be assigned to any of the geometries described by Gleiss, Norman & Wilson (2011) because of uncertainties with delta limited data points (Fig. 5D). All 15 dive profiles indicate that whale sharks quickly dived from the epipelagic zone to the maximum mesopelagic depth and back up into surface waters after the animal spent little (e.g., V-dives) or longer time periods (U-dives) at depth (Figs. 5A–5D). The pattern of rapid descents and ascents can also be inferred from the only two dive profiles that contain actual depth readings in the bathypelagic zone, but could not be reconstructed with confidence because of missing archived data points and most depth values being delta limited (Figs. 5E and 5F; see Brunnschweiler (2014) for details). Mean dive depth was negatively correlated with chlorophyll a concentration (Spearman Rank correlation; rs =  − 0.16, p <0.05).

Figure 5 Dive profiles.

(A) Isolated mesopelagic (green) V-dive performed by whale shark AF1 on 13 June 2009, 06:30–07:00; (B) two mesopelagic V-dives in series performed by whale shark JF1 on 3 April 2010, 12:45–14:45; (C) mesopelagic U-dive performed by whale shark JF1 on 7 May 2010, 13:45–18:45; examples of a (D) mesopelagic and (E and F) the only two bathypelagic (red) dive profiles that could not be assigned with confidence to any of the diving patterns described in Gleiss, Norman & Wilson (2011) due to missing archived data points (E and F), and delta limited dives and ascents (grey dots in (C–F); the whale shark was deeper (downward-pointing arrows) or less deep (upward-pointing arrows) than the archived value indicates; see Brunnschweiler (2014) for details).

Discussion

Juvenile and adult whale sharks displayed clear differences in their movement patterns. The five adult whale sharks tracked from the GoC, which included four adult females that were visually assessed to be pregnant, all moved a significant distance following tagging and spent the majority of their time in the open ocean. The females moved south and then to the north, offshore of the peninsula of Baja California, whereas the only male tracked moved straight to the south. In contrast, based on tracks and photo-identification data, juveniles showed a high degree of site fidelity to the GoC. Strong thermal gradients were positively associated with whale shark occurrence, indicating that they were spending time in frontal zones that were associated with upwelling systems. Meso- and bathypelagic dives were regularly recorded from all sharks.

Size- and sex-based segregation of whale sharks has been well-documented previously within the GoC area. This can be summarized as an offshore and coastal division, with adult females typically found in oceanic waters around or outside the 200 m isobath, while juveniles are found in coastal areas (Ramírez-Macías, Vázquez-Haikin & Vázquez-Juárez, 2012). Adults, specifically pregnant females, are seasonally present at Banco Gordo and Espíritu Santo Island (Eckert & Stewart, 2001; Ramírez-Macías et al., 2007; Ramírez-Macías, Vázquez-Haikin & Vázquez-Juárez, 2012; Ketchum, Galván-Magaña & Klimley, 2013). Juvenile whale sharks have been routinely observed feeding on calanoid copepods in shallow waters off Bahía de Los Angeles (Nelson & Eckert, 2007), and separately at Bahía de La Paz (Clark & Nelson, 1997; Hacohen-Domené, Galván-Magaña & Ketchum-Mejia, 2006). Long-term photo-ID from these areas, which have significant connectivity between them, found that the sharks present were typically juveniles, and predominantly males (Ramírez-Macías, Vázquez-Haikin & Vázquez-Juárez, 2012). Short-term residency periods of up to 153 days were noted for individual sharks, with a high level of inter-annual resightings (Ramírez-Macías, Vázquez-Haikin & Vázquez-Juárez, 2012). In the present study, all six juvenile whale sharks tagged at Bahía de La Paz between 2008 and 2010 were resighted several times in the GoC, both with and without tags, after days to months of tracking.

Adult sharks, particularly adult females, are conspicuous by their absence in most coastal aggregations (Rowat & Brooks, 2012; Hueter, Tyminski & De la Parra, 2013; Rohner et al., 2015b). Given the number of researchers surveying coastal areas around the world, the possibility that large whale sharks are commonly using this habitat seems remote (Hueter, Tyminski & De la Parra, 2013). Dedicated surveys in some regions (Ramírez-Macías, Vázquez-Haikin & Vázquez-Juárez, 2012; Ketchum, Galván-Magaña & Klimley, 2013), and observations from others (Afonso, McGinty & Machete, 2014; Robinson et al., 2016), indicate that mature whale sharks primarily inhabit offshore habitats. While the use of inshore habitats by juvenile sharks has been associated with zooplankton biomass in Bahía de La Paz and Bahía de Los Angeles (Hacohen-Domené, Galván-Magaña & Ketchum-Mejia, 2006; Nelson & Eckert, 2007; Ketchum, Galván-Magaña & Klimley, 2013), the seasonal biomass of zooplankton was not strongly correlated with adult shark presence in the outer Bahía de La Paz (Ketchum, Galván-Magaña & Klimley, 2013). This defined segregation between adult (female) and juvenile (largely male) sharks may therefore relate to differences in behavioural strategy, possibly including a dietary shift. Based on our results, adult female whale sharks appear likely to be found associated with offshore frontal systems. This supports recent studies from the Galapagos Islands (Hearn et al., 2016) that obtained detailed movement data from adult female sharks tagged at Darwin Island in the north of the archipelago. Many of these sharks travelled westwards, correlated with the flow of the South Equatorial Current, and close to the Equatorial Front which is a known biologically productive area (Hearn et al., 2016). Planktivorous basking shark Cetorhinus maximus sightings have also been strongly associated with tidal and large-scale frontal systems in the Eastern Atlantic (Sims, 2008; Miller et al., 2015).

The regular presence of female sharks, almost exclusively pregnant, in deep waters in the southern part of the GoC suggests that their presence is related to breeding. While the gestation period of whale sharks is not known, the pregnant female tagged in May 2012 at Banco Gordo, was also pregnant when observed in the Revillagigedo Islands in November 2012. Another pregnant female was photographed at the Revillagigedo Islands in November 2012 and resighted in May 2013 at Banco Gordo (D Ramírez-Macías, 2013, unpublished data). Therefore, it is possible that pregnancy lasts seven months or longer, if the observed distensions of the abdomen in each individual whale shark were pregnancies with the same litters. Individuals are transient to areas such as Banco Gordo (Ramírez-Macías, Vázquez-Haikin & Vázquez-Juárez, 2012; Ketchum, Galván-Magaña & Klimley, 2013) and, over the duration of tracking four pregnant females in this study (44–134 days), and three pregnant females tracked for 30–665 days by Eckert & Stewart (2001), all spent the majority of their time offshore from the GoC. While the small sample size limits interpretation, it is possible that parturition takes place in this offshore region. A very small, free-swimming whale shark pup was sighted at Espíritu Santo Island on 4 July 2015 (Fig. S3B). Oceanic pupping has been hypothesized for Atlantic whale sharks (Hueter, Tyminski & De la Parra, 2013), which is also supported by preliminary stable isotope profiles that suggest a transition from a pelagic offshore life of smaller (<4 m) whale sharks to a relatively more coastal habitat as size increased in whale sharks caught from India (Borrell et al., 2011). Photo-identification data from 2003 to 2014 has documented one female returning to Banco Gordo after seven years; the female was pregnant on both occasions (D Ramírez-Macías, 2013, unpublished data). Continued research effort in this area may be rewarded with further inter-annual resightings of pregnant sharks, providing valuable data on reproductive periodicity, and detailed documentation of the ontogenetic habitat shift that may be a feature of this population.

Adult female whale sharks moved into the Southern California Pacific, and the single male into the Mexican Pacific Transition. Both areas are deep eco-regions with offshore islands and seamounts (Wilkinson et al., 2009) that appear to be favoured habitats by adult whale sharks and other predators (Croll et al., 2012; Hearn et al., 2013; Afonso, McGinty & Machete, 2014). Whale shark neonate habitat is poorly known, although they have typically been found in or close to oceanic waters (Rowat et al., 2008). Neonatal pups are thought to have limited swimming ability, so the presence of both pregnant females and very small (∼2 m) juveniles around Bahía de La Paz, coupled with genetic evidence for natal philopatry to the GoC, suggests that pupping takes place nearby (Ramírez-Macías et al., 2007). While the oceanic area off the GoC is a noted hotspot for predatory fishes (Block et al., 2011), coastal waters within the GoC have a low observed density of potential whale shark predators and may be used as a refuge (Ramírez-Macías, Vázquez-Haikin & Vázquez-Juárez, 2012).

Although the adult sharks spent their time offshore over the period of tracking, largely in bathymetrically non-constraining habitat, all the sharks spent most of their time in the top 6 m of water, experiencing temperatures of 20–26 °C. Though the six sharks that left the tagging area did dive to meso- or bathypelagic depths, they spent a relatively small amount of time there. Brunnschweiler & Sims (2012) hypothesized that dives into the meso- and bathypelagic zones are related to foraging (searching) behaviour, and thus more extensive dives are expected to be more likely to occur in less productive deep oceanic waters. Interpretation is, however, complicated by the fact that sharks may be responding to deeper prey patches that are undetectable by remote sensing methods (Schick et al., 2013). While the observed deeper (V) dives may be linked to searching behaviour, diving can also reduce the cost of travel by whale sharks (Gleiss, Norman & Wilson, 2011), which would be of similar benefit during movement through less productive waters. Deep dives typically occurred in either the early morning or late afternoon. This pattern has also been observed in whale sharks from the Atlantic (Tyminski et al., 2015) and Western Australia (Wilson et al., 2006). Wilson et al. (2006) suggested that these crepuscular dives may be a means for a visual predator to exploit vertically-migrating prey during a short window of vulnerability. Geomagnetic navigation is also a possible driver, with the sharks plausibly diving to obtain a better ‘fix’ at around dawn and dusk (Tyminski et al., 2015) when magnetic intensity reaches its highest values (Willis et al., 2009).

All except one shark displayed a reverse diel vertical migration (rDVM; dawn ascent, dusk descent) overall, with deeper mean depth during the hours of darkness. However, 72.7% of all mesopelagic or deeper dives took place during the day. Little is known of the foraging ecology of adult whale sharks in the Eastern Pacific, or of juvenile sharks away from coastal areas of the GoC. Shark AM1, which had the greatest mean depth and number of dives into the meso- and bathypelagic zones of all the sharks in this study, was the only mature male that was tracked, further indicating the possibility of sex-specific differences in habitat use. The shallow mean depth of most sharks, and orientation towards frontal systems, suggests that surface prey represent an important food source. However, observer-independent dietary studies of southern African whale sharks suggest that they will also forage on deep-water zooplankton and fishes in oceanic waters (Rohner et al., 2013), which corresponds with observations of meso- and bathypelagic diving in that area (Brunnschweiler & Sims, 2012) that are comparable to those observed in the present study. Similar fatty acid studies on GoC sharks may provide insight on their motivations underlying vertical movements in this region.

While breeding is likely to occur in the GoC region, few adult male whale sharks have been observed during long-term field studies there (Ramírez-Macías, Vázquez-Haikin & Vázquez-Juárez, 2012). This was also the case at Darwin Island in the Galapagos (Acuña Marrero et al., 2014). While genetic results indicate that there is some connectivity between whale shark aggregations within the Indo-Pacific (Vignaud et al., 2014), to date there have been no photographic resightings of GoC whale sharks in other areas in the Eastern Pacific, such as Cocos Island in Costa Rica (White et al., 2015) or the Galapagos Islands in Ecuador (Acuña Marrero et al., 2014). While one of the tags deployed on a juvenile shark by Eckert & Stewart (2001) is thought to have travelled around 13,000 km across the Pacific, it is now considered unlikely that the tag was still attached to that shark (Brunnschweiler et al., 2009).

Conclusion

Regular inter-annual resightings of juvenile sharks in the GoC (Ramírez-Macías, Vázquez-Haikin & Vázquez-Juárez, 2012), coupled with the short-term residency displayed by tagged sharks in this study, suggest that juvenile whale sharks display site fidelity to these coastal waters, at least over the period they remained tracked. While adults may demonstrate some reproductive philopatry (Ramírez-Macías et al., 2007), they appear to be considerably more mobile and primarily oceanic. Conservation efforts for the species should therefore focus on both local scales, where anthropogenic threats to coastal feeding areas could have a disproportionate impact at a population level (Ramírez-Macías, Vázquez-Haikin & Vázquez-Juárez, 2012), as well as regional threats to breeding populations. Specifically, of the five sites that were regularly used by whale sharks in this study, two—Bahía de La Paz and Banco Gordo—remain unprotected. Our tracking data, together with long-term photo-ID data, demonstrates connectivity between all five sites. Improved protection should be investigated as a potential regional conservation measure, given the international significance of the GoC and surrounding regions to whale shark reproductive ecology.

Supplemental Information

Figure S1 Adult female whale shark and whale shark pup sighted at Espiritu Santo Island

(A) Adult female whale shark WS7 photographed on the day of tagging at Espiritu Santo Island (Table 1, Fig. 1A); (B) whale shark pup sighted at Espiritu Santo Island on 4 July 2015. Photographs copyright Carlos Aguilera Carderón (A) and Jose Maria Urbalejo Calvillo (B).

Click here for additional data file.

Figure S2 Resampled locations (point density data)

Along previously reported tag-specific longitudinal and latitudinal Gaussian error fields (0.16ºin longitude and 1.19ºin latitude).

Click here for additional data file.

Figure S3 Average point density data and environmental data

Averaged (0.5ºgrid) point density data (A) and environmental data (B–D); B –sea surface temperature (ºC); C –sea surface temperature gradients (ºC/10 km); D –chlorophyll a concentration. Note: these were the input data for the GLMM model.

Click here for additional data file.

Data S1 Raw Data

Archived depth, temperature, and latitude and longitude data from original data reports from tags WS1 to WS8 received from Microwave Telemetry

Click here for additional data file.

The authors would like to thank Ricardo Vázquez-Juárez for the support in obtaining research authorisations and Alex Antoniou for his advice in tagging whale sharks. Field work was suported by Erick Higuera, Siddharta Velazquez, Paul Ahuja, Felipe Morales, Carlos Aguilera, Paulina Godoy and Alfredo Barroso. The authors thank Christoph Rohner for his comments on the manuscript.

Additional Information and Declarations

Competing Interests

Author Contributions

Animal Ethics

Field Study Permissions

Data Availability

The authors declare there are no competing interests.

Dení Ramírez-Macías conceived and designed the experiments, performed the experiments, wrote the paper.

Nuno Queiroz and Juerg M. Brunnschweiler analyzed the data, wrote the paper, prepared figures and/or tables, reviewed drafts of the paper.

Simon J. Pierce wrote the paper.

Nicolas E. Humphries and David W. Sims contributed reagents/materials/analysis tools, wrote the paper.

The following information was supplied relating to ethical approvals (i.e., approving body and any reference numbers):

This research was carried out under the general auspices of CONACYT (Consejo Nacional de Ciencia y Tecnología), this is the relevant Mexican authority governing all research actions. Register numbers 030 (currently 1602199) and 13920.

The following information was supplied relating to field study approvals (i.e., approving body and any reference numbers):

This research was carried out under the general auspices of DGVS (Dirección General de Vida Silvestre), SEMARNAT (Secretaría del Medio Ambiente y Recursos Naturales) and CONANP (Comisión Natural de Áreas Naturales Protegidas). These are the relevant Mexican authorities governing all research actions on wildlife and protected animals and areas in Mexico. DGVS authorization numbers are: SGPA/DGVS/02677/08, SGPA/DGVS/02888/09, SGPA/DGVS/03848/10, SGPA/DGVS/03155/11, SGPA/DGVS/03362/12.

The following information was supplied regarding data availability:

The raw data has been supplied as Supplementary File.

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
