# Peer review of "Oceanic adults, coastal juveniles: tracking the habitat use of whale sharks off the Pacific coast of Mexico"

_PeerJ, doi:10.7717/peerj.3271_

## Round 0.1 · original submission · Major Revisions

This paper provides interesting and useful information about the movements and dive behavior of whale sharks. Having read the manuscript and assessing the input of three reviewers I am returning the manuscript for major revisions. It is currently not acceptable for publication in PeerJ.

Two reviewers indicated that the methods used to assess the movements of the animals required significant clarification, and in particular stressed the use of kernel density estimators with barriers, and the limitations of light-based location tags along and the uncertainly associated with positions generated. Reviewer 1 suggested that the process/software for determining positions should be more open, which fits well with the open-access philosophy of PeerJ. Authors should pay attention to the data sharing policy of PeerJ in this regard as well. Reviewers also indicated that the classification of sharks as pregnant, and how that relates to the conclusions of the paper, is in need of some attention.

Reviewer 1 provides excellent detail on needed revisions to the narrative, as well as to the figures and tables. These corrections should all be addressed in any subsequent submissions.

·

Basic reporting

Line 51: Sentence does not substantial improve support, please amend for context or remove.

Line 52: Sentence does not sufficiently explain the desire for such a study and does not provide sufficient justification beyond the data was collect. Please amend to reflect a central focus and need for the study.

Paragraph beginning on Line 94: This appears to be something that would be best included in the acknowledgements.

Line 114: Please state the minimum length that a juvenile male can be classified or an estimate

Figure 1: should be deleted

Figure 2: should be turned into a multipanel with the original Figure 2 and another panel of the inset area of Bahia de La Paz at a scale necessary to see tracks of WS1 and WS4.

Figure 2: Missing tracks from WS1 and WS4.

Figure 2 [legend]: It is unclear if this reflects the most probable track as denoted by the Methods or if this is the true geolocations. Are resights included in the tracks? If not, why not?

Figure 3: Choose a color scale that is more easily viewed by color-blind individuals

Figure 4: Choose a color scale that is more easily viewed by color-blind individuals. Change the dot colors as they can be seen against a purple background. Add what panel D depicts or add the label. It is unclear which track is WS2 and WS3, fix. Readers should not have to refer to other figures or tables to ascertain.

Figure 5: Panels B and D should be boxplots as they are individual data. See comment on Table 2.

Figure 7: Please state that dark grey was used for delta limited dives.

Table 2: Figure 5 could be summarized into an additional row for average. The Mann-Whitney U test was not explained in the Methods. Add. Maximum what?

Table 2 [legend]: Legend does not sufficiently explain what is going on in the table. Especially in relation the Mann-Whitney U test.



General Comments:
17 sharks were tagged but data related to the spatial ecology was only presented for 8. This was confusing. As the majority of the study was focused on horizontal movements and vertical movements, data from nonreporting tags seems fairly useless. In fact, the reading of the methods is encumbered by their inclusion. First, a decision to include the data or not needs to be addressed and, second, what happened with the nonreporting tags needs to be addressed, and, third, the writing needs to be clarified where the 17 tag data is reported.

The figures (and probably some analysis) seem to be produced in R. Please state that this software was used, as it is open-source.

Experimental design

Line 163: “a proprietary algorithm” – this impedes reproducibility; please state who owns the algorithm and what software it is available in.

Line 164: “custom-made software” – this impedes reproducibility; please state who owns the software or what it is licensed under or where it is available, and what it does.

Line 166: what was the criterion for the swim-speed filter? what data was used?

Line 176-181: All possible tracks or points should be weighted by the scoring (similar to ΔAIC) then incorporated into the kernel density estimates. Assuming the most probable track is the “true” path and then using it to estimate a kernel for the animals is inappropriate. Doing such underweights areas that are likely nearly as probable as the most probable path. Fix or demonstrate that other probable paths are so unprobable that they can be discounted. Additionally, using the default kernel density estimator in ArcGIS does not account for barriers. Please use kernel density accounting for barriers. As it stands, despite being clipped from the map, most of your kernel is on the tip of Baja California. Laffman & Taylor (2013) document one methodology that could do this.

Line 177: Please state which version.

Line 181: State which data.

Line 182: Spearman Rank correlation returns a significance value based on the assumption that the degrees of freedom is equal to the sample size. Was this corrected for as you have repeated measures? Fix, if not.

Line 206: See above on Spearman Rank correlation

Line 208: Please state the temporal scale used to create the mean chlorophyll-a concentration.

General Comments:
Mann-Whitney U tests requires that both groups (i.e. day versus night) are independent of one another. Observations from a given whale shark are not, unless otherwise demonstrated.

Validity of the findings

The conclusions are based on inappropriate use of statistical tests. These need to be addressed.

My biggest concern with validity beyond statistical tests, is that the longest DAL of juveniles is 58 days and only 2 tagged juveniles reported in the same year. This begs the question, how can these individual results be scaled up to local, and higher, population level patterns. Particularly, when the longest tag duration moved considerably towards the end of the deployment. It seems, that juveniles and adults (pregnant or otherwise) can only be compared within a year, as you show that movement can be quite different depending on the local chlorophyll-a distribution. Thus choices available to the population in one year are not equivalent to the next. It that case, you can compare 2 juveniles with 2 pregnant females and 1 adult in 2010. The other years are not truly useful for comparison in this light.
This in itself may be generous, as juveniles in 2010 were tagged in March and the tags had popped-off before the any other reporting animals were tagged. I believe these issues, at the very least, warrant considerable discussion. Additionally, further justification is necessary to support off-year or off-season comparisons made in the manuscript.

Reviewer 2 ·

Basic reporting

English is good and concise.

A few referencing issues, please review.

Methodology could be improved.

Maps need scales and labelling on some needs improving (Fig 2 is missing some sharks)...

Conclusions are mixed, chl a analysis done but states foraging at depth? Maybe some introduction about chl a and how the authors think it is linked to whale shark movements in the introduction? For me the offshore habitat looks like there is consistent low level production.

Experimental design

1.) There are no areas of confidence around the locations. Light Level data should not (in my opinion) be used to assess small-scale movements such as with Chl a. A tag with a known error radius should be used. It is known the confidence areas of light level data (even after improvements) are large; these should be shown on the images. Tag models need to be chosen carefully and their limitations understood.

2.) A KDE with barriers should be used (there is land in the current KDE) and then create 50 and 95% PVC for adult and juvenile sharks to make a comparison.

3.) Define onshore and offshore habitats for this study.

4.) Methodology is confusing (126-138)

5.) Chl a, where from? What resolution? Chl a is not a measure of productivity and if the sharks forage at depth then why use it? There is also a lag between Chl a and zooplankton.

6.) I suggest changing the names from WS1, two etc. to MF1 (Mature Female 1), MM1, JM1, etc. as it becomes confusing.

7.) 210-211 - tag detached from the shark and then started to transmit after the constant depth trigger?

8.) 219-232 - could be a table or map.

9.) Pregnancy - Although gestation is not known (but it is for other related species), it is known that they gestate up to 300 pups at the same time at different stages of development and therefore must give birth at different times, so it is likely that a female will remain pregnant for some time, if not permanently throughout adulthood. It is known they don't give birth to all pups at one time so it would not be surprising that she would still be pregnant after this small duration of time. They have the ability to store sperm, and it is not known how long they continue to fertilise eggs for.

10.) The tags are too few and the retention time is not long enough to.

11.) Sample size is not large enough for speculation about birth; the male moved offshore too?

Validity of the findings

See above

Additional comments

A nice attempt at a paper that will add to what is known about the whale shark around the GoC. In my opinion there are some fundamental issues with the data analysis and capabilities of the tags used that I have attempted to address. The tags are too few, retention time to short and sample size of each demographic too few to make the conclusions that are made. I suggest looking again at the strength of the data from the tags and working with that.

·

Basic reporting

No Comments

Experimental design

No Comments

Validity of the findings

See general comments to authors

Additional comments

This article is one of the first attempts to elucidate ontogenetic shifts in habitat use and behavior of whale sharks, a long overdue exercise due mainly to the lack of individuals other than immature males at most coastal aggregation sites. I recommend it for publication with minor revisions and a few thoughts that the authors might like to consider in their discussion.

There are a couple of minor issues with some of the figures:
In Fig.4 it is hard to distinguish the tracks between WS2 and WS3, and the caption for 4d is missing (actually I think it is there but not labeled).
In Fig 5 the IDs of the whale sharks do not correspond to what is in the text and table. The bar charts are a little confusing – why not also split them by age group (adult vs juvenile, and use boxplots instead – especially for the thermal use: what is their thermal niche in relation to the available thermal regime in the area they are in?)

It looks like mean overall depth is greater at night, yet the deepest dives are actually in the morning (and perhaps at dusk too?). I feel that the authors might want to explore that a little more – it is unclear what is meant by “exploratory behavior”. Could it be some navigational aid?

On line 369-70, a shift in diet might also explain this.
On line 400-02, I would feel more comfortable with “suggests” rather than “indicates” as this is quite speculative at this stage. Not that I disagree with what is being said! But it is unusual that if pupping occurs nearby, so few records of neonates have been made.
Line 439 – why do the authors think that breeding is likely to occur in the GoC?

On the use of the term “pregnant”. The authors use a reference (Ramirez et al 2012) to support their claim that these large females are pregnant, yet in that article the authors simply use a visual cue (distended belly) to assign pregnancy. It would be difficult to prove in the field, and I agree that it is more than likely that they are pregnant, but having had a similar issue in my own publications regarding “pregnant” whale sharks, I feel that at the very least, the authors should comment on this in their discussion. I am not however suggesting that they remove the term… only qualify it somewhat.

---

## Round 0.2 · accepted · Accept

After reading the revised manuscript and final comments from Reviewer 1, I feel this paper is now ready for publication in PeerJ. The authors have done a great job addressing concerns with the initial manuscript and it is greatly improved.

·

Basic reporting

You might wish to move Table 3 into the text to remove it. It does not contain a ton of information and moving it into text might allow for a better looking publication as well as streamline reading the results.

Experimental design

I applaud the authors for their revision. It is considerably improved. I appreciated the clarity in which the methods were stated and applied.

I still, respectfully, disagree with the application of the Mann-Whitney U test but I do not feel it hinders the manuscript. Further, debate on its application is not productive via review.

My only concern with the methods is the normal distribution with the GLMM. While the PQL does help with the non-normal data, I'm a bit curious why a normal likelihood was chosen when the data was most likely log-normal (visually guesstimated from Figure S3; which was a nice addition). Depending on the other reviewers or in the future, this might be something to try; especially given the very close p-value for Chlorophyll-a, moving to a different likelihood structure might make it significant. Just a thought.

Validity of the findings

The revision has substantially improved the manuscript and I have no comments to improve. The authors are careful with their language and the importance of the results are communicated well.

Additional comments

I greatly appreciate the authors making the revision that they did. Often, it can be trying as a reviewer to make comments and to not see any improvement. The consideration of reviewer comments as well as the authors' careful analysis came through in the revision.